# Recurring marine phosphorus spikes during major palaeozoic mass extinctions and climate change

**Matthew S. Dodd** [1,2,3,13] ✉, **Chao Li** [2,3,13] ✉, **Zihu Zhang**[2,3], **Aleksey Y. Sadekov**[1], **André Desrochers** [4,5], **Olle Hints** [6], **Detian Yan**[7], **Xiangrong Yang**[7,8], **Annette D. George**[1], **Maya Elrick**[9], **David White**[9], **Wenkun Qie** [10], **Bo Chen**[10], **Andrew S. Merdith** [11] & **Benjamin J. W. Mills** [12]

Mass extinctions in the early Palaeozoic have been attributed to global climate change and ocean anoxia with elevated phosphorus (P) proposed as a key driver. However, this hypothesis has lacked geochemical support due to the absence of proxies that can reconstruct changes in marine P availability. Focusing on the Late Ordovician Mass Extinction (LOME) and the Late Devonian Mass Extinction (LDME), we present carbonate-associated phosphate (CAP) data from seven globally distributed sections, providing a proxy record for seawater P variation across these events. Our data reveal short-lived, globally coherent P pulses that coincided with both events. These transient P surges align with biodiversity loss, widespread anoxia, and seawater temperature declines, suggesting a link between P flux, ocean anoxia, and global climate shifts, as supported by biogeochemical model results. These findings provide an empirical connection between brief marine P pulses and ecological crises during the LOME and LDME.

The Late Ordovician and Late Devonian Mass Extinctions (LOME and LDME) occurred around 445 and 372 million years ago (Myr), respectively. These two oldest of the "big five" extinction events of the last 500 million years, wiped out ~85 and 80% of marine fauna, respectively[1,2]. These two extinction events differ from the remaining "big five" extinctions in that they correlate with global cooling, as opposed to global warming[3], yet they may still have been linked to large igneous provinces and associated volcanic activity[4]. They are both associated with global cooling events with greater than 5 °C decreases in seawater temperatures within the tropics[5,6], along with oscillating extents of ocean anoxia[7,8]. Both events are traditionally viewed as having at least two extinction pulses (LOME 1 and 2, and the Lower and Upper Kellwasser (LKW and UKW) events of the LDME).

The primary cause(s) of these events is (are) debated; however, hypotheses involve enhanced terrestrial nutrient influx related to either large igneous province emplacement[9] or the proliferation of land plant-related chemical weathering (especially during the LDME)[10]. In either scenario, elevated P flux to the oceans leads to enhanced

[1]School of Earth and Oceans, University of Western Australia, Perth, WA, Australia. [2]State Key Laboratory of Oil and Gas Reservoir Geology and Exploitation & Institute of Sedimentary Geology, Chengdu University of Technology, Chengdu, China. [3]International Center for Sedimentary Geochemistry and Biogeochemistry Research, Chengdu University of Technology, Chengdu, China. [4]Earth and Environmental Sciences, University of Ottawa, Ottawa, ON, Canada. [5]Société du patrimoine mondial Anticosti, Port-Menier, Québec G0G 2Y0, Canada. [6]Department of Geology, Tallinn University of Technology, Ehitajate tee 5, 19086 Tallinn, Estonia. [7]Key Laboratory of Tectonics and Petroleum Resources of Ministry of Education, China University of Geosciences, Wuhan, China. [8]School of Geosciences, Yangtze University, Wuhan, China. [9]Earth and Planetary Sciences, University of New Mexico, Albuquerque, NM, USA. [10]State Key Laboratory of Palaeobiology and Stratigraphy, Nanjing Institute of Geology and Palaeontology, Chinese Academy of Sciences, Nanjing, China. [11]School of Physics, Chemistry and Earth Sciences, Adelaide University, Adelaide, SA, Australia. [12]School of Earth and Environment, University of Leeds, Leeds, UK. [13]These authors contributed equally: Matthew S. Dodd, Chao Li. ✉e-mail: matthew.dodd@uwa.edu.au; chaoli@cdut.edu.cn

marine productivity, oceanic anoxia and carbon burial, followed by atmospheric $CO_2$ drawdowns and global temperature decline[10–16]. This conceptual model is widely cited and often assumed, yet to date there is no geochemical evidence that records changes in seawater phosphate concentration through these events. Previous support has largely come from burial-based inferences (e.g., P speciation[17]) or supply-side indicators (weathering proxies[18]), which do not measure seawater phosphate and cannot reconstruct its temporal variability. As a result, the proposed role of phosphorus in these extinctions remains speculative and poorly constrained.

In the ocean, phosphorus is partitioned among dissolved inorganic phosphate, dissolved and particulate organic P, and inorganic phases such as Fe-oxide–bound and detrital or authigenic phosphate minerals. Biological uptake converts dissolved phosphate into organic P, which is rapidly remineralised and hydrolysed back to dissolved phosphate on timescales much shorter than ocean mixing, so that the size of the seawater P reservoir is tightly coupled to processes in underlying sediments[19]. Within sediments, microbial degradation of organic matter and reductive dissolution of Fe oxides release both organic and Fe-bound P to porewaters; this regenerated P can either be trapped as authigenic phosphate minerals or diffuse into bottom waters, with anoxic – especially euxinic – conditions strongly favouring return of P to the water column rather than burial[20,21]. Consequently, episodes of high productivity and expanded seafloor anoxia tend to elevate seawater P concentrations by enhancing fluxes of P pools from the sediments to the water column, relative to periods of reduced seafloor anoxia, which results in more P being retained by sediments[19,22].

In this study, we applied carbonate-associated phosphate (CAP), a recently developed proxy that can reconstruct changes in seawater P availability, to seven globally distributed locations across the LOME and LDME. This allows us to reconstruct changes in marine P concentration across these extinction intervals and evaluate hypotheses of P-driven environmental change and mass extinction.

## Results
### Late Ordovician Mass Extinction
Four different sections were studied for the LOME, including the Pointe Laframboise section and western Ellis Bay section from Anticosti Island, Canada; the Viki drill core section, Estonia; and the Wukemuchang section, South China (Fig. 1a–e; see Supplementary Information 1 for section details). All sections preserve positive $\delta^{13}C_{carb}$ excursions, which are hallmarks of the LOME, but with varying magnitudes (1–5‰), and depositional ages. The age date variability of the $\delta^{13}C_{carb}$ excursions has been recognised globally[23–26]. Consequently, graptolite and chitinozoan biostratigraphy is used to correlate geochemical trends between sections. Among these sections, bulk rock CAP extracts begin to slightly increase from values of around 0.06 mmol/mol or 0.025 mmol/mol in the case of the Viki drill core samples, after the *Normalograptus extraordinarius* Zone, following the first extinction pulse (LOME 1, Figs. 1a–e and 2). Then, CAP values peak around 0.12-0.18 mmol/mol in all sections at the base of the *Normalograptus persculptus* Zone, or the temporally equivalent *Conochitina scabra* Chitinozoan Zone in Estonia[27,28]. This corresponds to the start of the second mass extinction pulse (i.e., LOME 2; Figs. 1b–e and 2a). Following peak values, bulk rock CAP values decline throughout the *N. persculptus* Zone, reaching minimum values of 0.02 to 0.14 mmol/mol in the early Silurian.

To assess the robustness of the bulk rock CAP data, we used in-situ laser ablation to sample and measure CAP values in petrographic thin sections from samples in Pointe Laframboise, Ellis Bay, and the Viki drill core sections (Fig. 1b, c and e; Supplementary Information Fig. 2–4). In all sections, CAP values from in-situ laser ablation are generally higher than bulk rock extracts, on average, by 30%, 50%, and 220% in the Pointe Laframboise, Ellis Bay, and the Viki drill core

sections, respectively. Despite differences in absolute values, the CAP trends from both bulk rock and in-situ laser ablation data show similar trends (see Supplementary Information 2 for further details).

### Late Devonian Mass Extinction
LDME sections include the Horse Spring Range drill core (Western Australia), Yangdi (South China) and Devils Gate (Nevada, USA) (Fig. 1f–i; see Supplementary Information 1 for section details). Except for the Yangdi section, all sections record two positive $\delta^{13}C_{carb}$ excursions of up to +3‰ to +5‰, whereas the Yangdi section only captures the youngest of the two $\delta^{13}C_{carb}$ excursions, which typically comprise the LDME. In the Devils Gate and Horse Spring Range sections, bulk rock CAP values begin to slightly increase at the base of the *Palmatolepis rhenana* conodont Zone from around 0.05 and 0.1 mmol/mol, prior to the first extinction pulse (i.e., LKW; Figs. 1g–i and 2b) to peak values of 0.09 and 0.12 mmol/mol, respectively, around the beginning of the *Palmatolepis linguiformis* conodont Zone, which correlates temporally with the second extinction event start (i.e., UKW; Fig. 1g–i). CAP trends in the Horse Spring Range section are less clearly defined owing to a large spread in data points, resulting in a more muted CAP signal through the LKW compared with the Devils Gate and Yangdi sections. In the Yangdi section, CAP values increase through the upper *P. rhenana* Zone from values of around 0.12 mmol/mol to peak values of around 0.2 mmol/mol at the start of *P. linguiformis* Zone. This matches the CAP trends in the other two sections. All sections then record a swift decline in CAP values through the *P. linguiformis* Zone. They reach minimum values of 0.06, 0.09, and 0.05 mmol/mol in the Horse Spring Range, Yangdi and Devils Gate sections, respectively, around the Frasnian-Famennian boundary, which broadly marks the end of the LDME (Figs. 1g–l and 2b). Subsequently, all sections show an increase in CAP values throughout the *Palmatolepis triangularis* Zone, reaching peak values of 0.13, 0.24, and 0.13 mmol/mol in the Horse Spring Range, Yangdi and Devils Gate sections, respectively.

## Discussion
### CAP Data Evaluation
Phosphorus in seawater is incorporated into carbonate minerals proportionally to the local seawater P concentration[29–31]. Accordingly, ancient marine phosphorus levels can be inferred from phosphate concentrations in carbonate minerals, provided that seawater chemistry (for example pH, alkalinity, and temperature) and carbonate mineralogy are also taken into account because they influence elemental uptake[29,30,32,33]. In marine sediments, P resides chiefly in phosphate minerals, but additional fractions occur in Fe-associated phases, on mineral surfaces, and within organic matter and carbonate minerals[21]. These P pools, with the exception of carbonate-bound P, are largely controlled by redox conditions and biological activity[20], and not directly attributable to dissolved seawater P concentrations[21]. In contrast, phosphate bound in carbonate minerals (i.e., CAP) can be more directly linked to seawater dissolved P concentration. In order to extract and measure CAP, a previously developed and validated leaching protocol was used (see Methods)[29,32] to reconstruct relative marine P variations across the LOME and LDME events while minimising contaminations from other P pools.

Estimates of relative changes in palaeoceanic P levels can be discerned from CAP if contamination, diagenetic alteration, and potential changes in oceanic chemistry can be correctly evaluated[29]. Our bulk rock CAP extractions show that contamination from other sedimentary P pools, was minimal because we observe little correlation between CAP and La (an element used for tracking phosphate mineral dissolution[29,34]) (Supplementary Information Table 1; Supplementary Information 3). Additionally, in-situ CAP analyses were screened during data processing to select only those analytical intervals with no covariation between P and the concentration profiles of Al, S, La and Fe, which indicate P contamination from detrital minerals (Al), organic

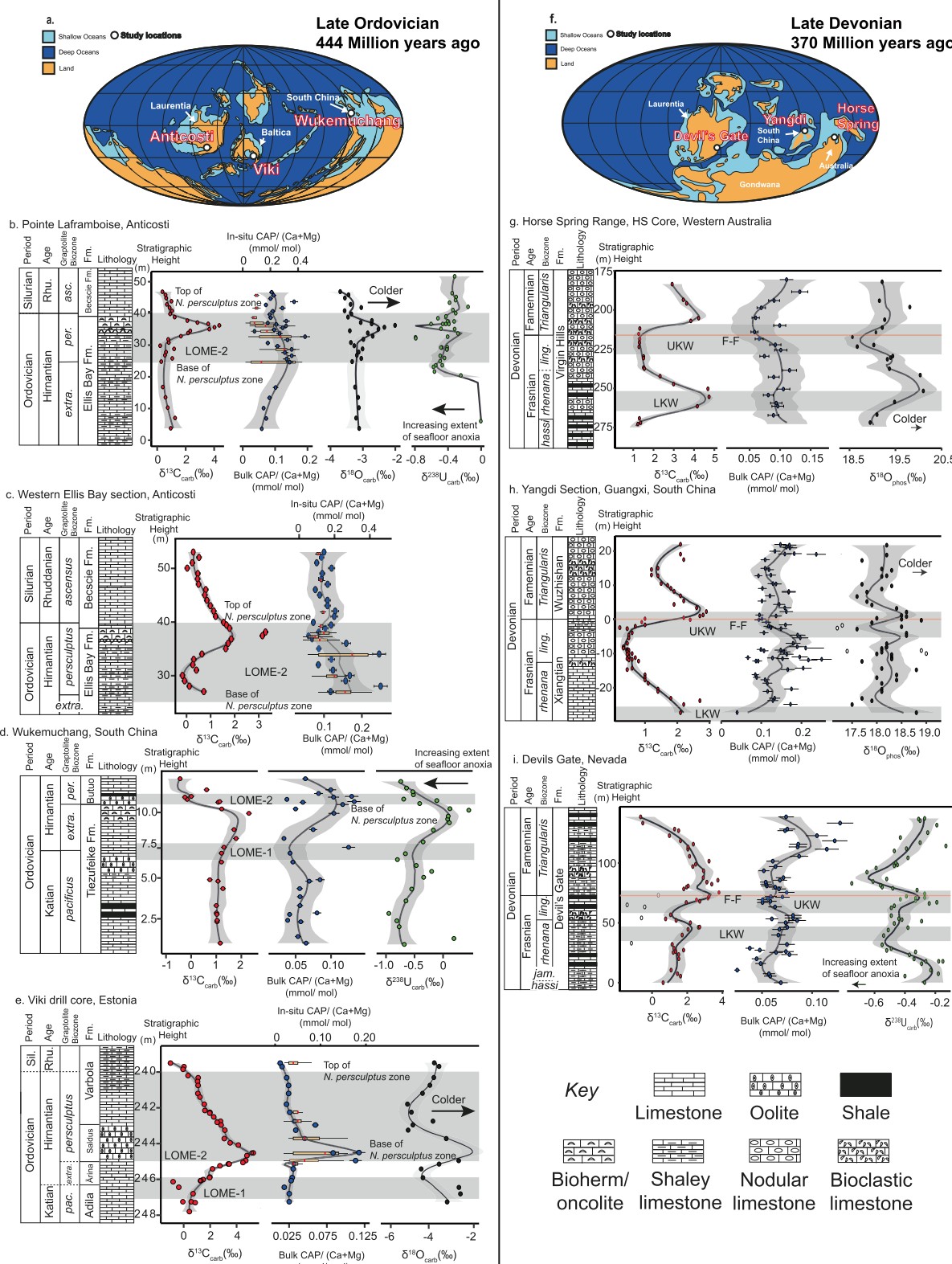

**Fig. 1 | Biogeochemical data from study sections during the late Ordovician and Devonian mass extinction events. a** Global paleogeographic reconstruction during the late Ordovician with study site locations modified after ref. 51. **b** Pointe Laframboise, Anticosti, Québec, Canada. **c** West side of Ellis Bay, Anticosti, Québec, Canada. **d** Wukemuchang, South China. **e** Viki drill core, Estonia. **f** Global paleogeographic reconstruction during the late Devonian, with study site locations modified after ref. 51. **g** Horse Spring drill core, Western Australia. **h** Yangdi, South

China. **i** Devils Gate, Nevada. Geochemical data sources: CAP (this study), C and O isotope data (Anticosti sections this study and partially for Viki drill core); all other O-, U-isotope data from refs. 6,7,17,32,38,39. In-situ CAP data is represented by box plots and bulk rock CAP by blue points. VPDB, Vienna Pee Dee Belemnite. Stratigraphic heights are unique to each section and do not follow any geological markers. Error bars show a 10% error margin. Black line through data = LOESS smoothing with grey shading representing the 95% confidence interval.

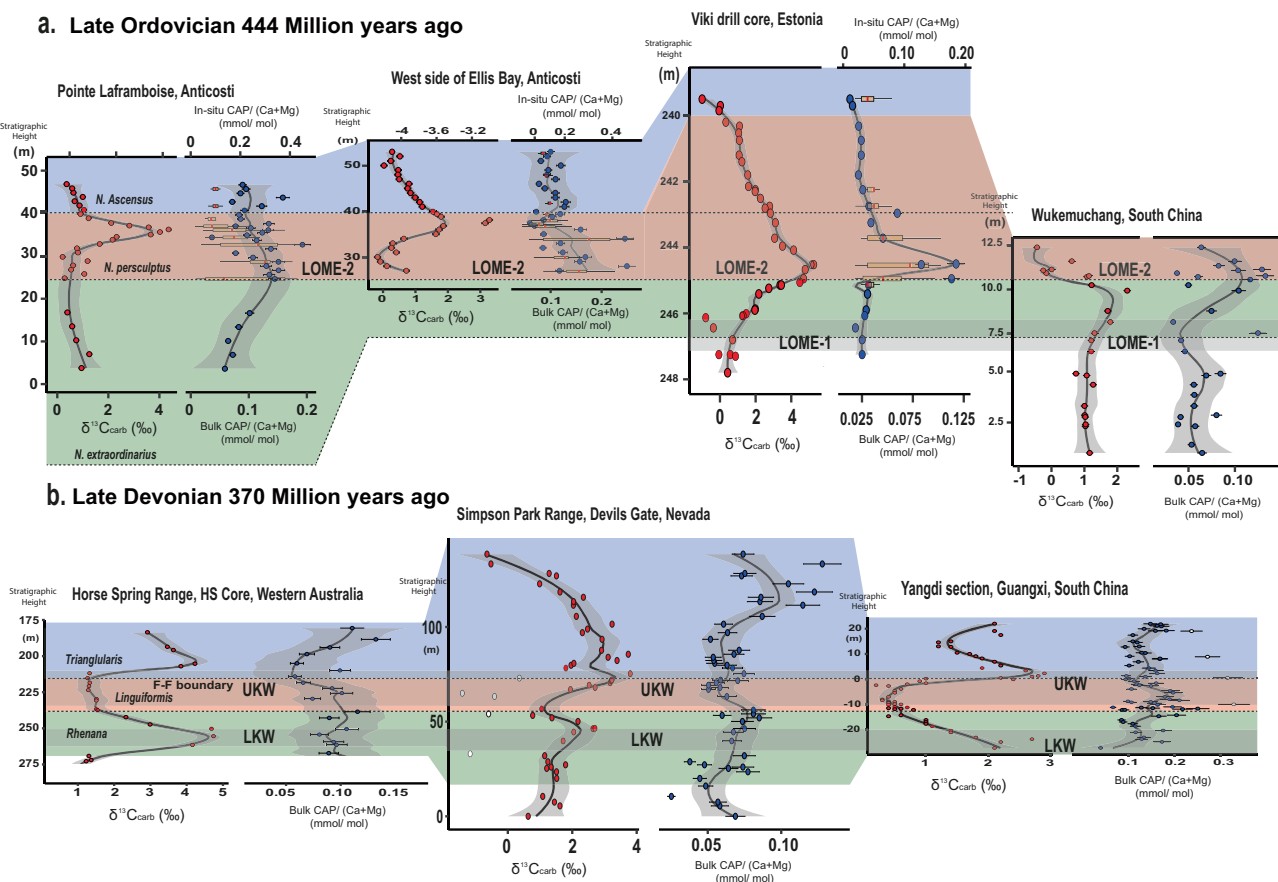

**Fig. 2 | Correlation of globally distributed sections based on biozones which recorded the Late Ordovician and Late Devonian Mass Extinctions (LOME and LDME). a** Late Ordovician sections with two extinction phases (LOME 1 and LOME 2) indicated. **b** Late Devonian sections with two extinction phases [the Lower and Upper Kellwasser (LKW and UKW) events]. Coloured areas correspond to different biozones. Note that CAP trends are temporally consistent among sections.

matter (S), phosphate (La), and iron-oxide (Fe) minerals, respectively. Reassuringly, both bulk rock and in-situ methodologies return similar CAP trends, albeit with some differences in absolute CAP values (Fig. 1; see Supplementary Information 2 for further details), confirming the reliability of our CAP trends.

The observed globally synchronous CAP trends during the LOME and LDME (Supplementary Information Fig. 1) pose a challenge to reconcile with local diagenetic alteration. However, globally synchronous diagenetic alteration of carbonate has been purported to result after shoreline regression exposes platforms to meteoric waters[35]. To assess the potential effects of local and widespread diagenetic alteration (in part due to global sea-level changes), we utilise trace elements and isotopic methods as described below (see Supplemental Information for more details).

The best evidence for the preservation of near-primary carbonate chemistry comes from the close agreement between seawater temperature estimates from conodont apatite[7,36] and carbonate oxygen isotope (Fig. 1b,e) palaeothermometers at the Anticosti and Viki sections. Both record a decline in temperature, with estimates within 20% of each other[6,7,36,37]. Given that carbonate oxygen isotopes are strongly susceptible to diagenetic alteration and are expected to be altered at orders-of-magnitude lower fluid-to-rock ratios than CAP[32], we can reasonably infer that CAP trends reflect near-primary chemistry at least in these sections. We note, however, that the oxygen isotope composition of carbonate from the Wukemuchang section is significantly depleted $\sim -9$‰ compared to that of carbonate from Anticosti and the Viki drill core samples[38]. Despite this evident diagenetic alteration of oxygen isotopes at Wukemuchang, CAP trends remain consistent

among all sections, potentially reflecting the more resistant nature of CAP preservation than oxygen isotopes in carbonate samples, and/or the low P content of meteoric waters that can lower $\delta^{18}O_{carb}$ values during alteration[39]. In contrast, coeval carbonate and conodont apatite $\delta^{18}O$ values from the LDME Horse Spring Range and Yangdi sections do not share similar temperature estimates, suggesting that the carbonate $\delta^{18}O$ values have undergone diagenetic alteration. Despite this, CAP trends remain consistent among the three globally distributed sections, in spite of variable bottom water redox conditions (Fig. 1g–i), which would strongly modify early local diagenetic processes. For example, in the Horse Spring Range section, bottom waters were oxygenated throughout the extinction events[40], whereas in the Devils Gate section, bottom waters were mostly anoxic throughout the extinction events[8]. This would significantly modify diagenetic cycling of phosphorus, with P being immobilised within the sediments as Fe-P complexes and phosphate minerals in oxygenated settings and released from sediments in anoxic settings. Moreover, no statistically significant correlations emerge between CAP values and carbonate oxygen isotopes or Sr and Mg in the LDME sections (Supplementary Information Table 1). These lines of evidence above support minimal diagenetic alteration of CAP in these samples.

Previous studies have shown that samples from the Viki drill core section contain late diagenetic carbonate cement that post-dates deposition by tens of millions of years[41]. We analysed these cements and carbonate clasts and found that CAP values are significantly lower in the cements compared to the clasts. This suggests that lower CAP values in these samples may have resulted from post-depositional processes. Despite this, the bulk rock CAP values in those samples with

post-depositional cement still record some of the highest CAP values from the Viki samples. This suggests that these diagenetic components did not affect the CAP trends.

Given that trace element impurities such as Sr, U, and Mg in carbonate gradually diminish during recrystallisation to more ordered forms due to exclusion recrystallisation[29,42], we consider carbonate materials carrying the highest Sr, U, and Mg concentrations as those preserving chemistry close to that of the primary precipitates[43]. Consequently, preservation effects may result in a positive correlation between CAP and trace elements like Mg and Sr. We do not find strong correlations between CAP and Sr concentrations from the in-situ CAP data, but we do find a strong correlation of CAP with Mg within individual thin sections from the Anticosti and Estonian sections (Supplementary Information Figs. 1 to 4). Correlations exist within individual samples, which may represent variable CAP preservation. However, the CAP trends from the entire sample set do not show a statistically significant correlation among Mg or Sr and CAP. This suggests that bulk rock CAP trends are unaffected by preservation bias, despite in-situ analysis revealing evidence for alteration of initial CAP value within individual samples (Supplementary Information Table 1).

What is particularly noteworthy is that CAP trends are replicable across sections despite widely varying carbonate lithology. For example, in the Viki drill core section, the CAP peak at the *N. persculptus* Zone occurs across a shift from micrite to oolite, whereas the same CAP peak is expressed across bioclastic limestone in the Anticosti sections. Such a disconnection between carbonate lithology types and CAP trends suggests that lithological and associated diagenetic changes are unlikely to be a major control on the observed CAP trends. Likewise, the Devils Gate and Yangdi sections remain comparable to one another despite CAP trends being expressed across alternating layers of shale and limestone at Devils Gate, compared to solely bioclastic limestone at Yangdi. In summation, a range of independent trace elements and isotopic data support a near-primary seawater origin of the CAP trends in the LOME and LDME sections, with evidence for limited diagenetic overprints.

If the CAP trends are mostly primary, they record local seawater P concentrations at the time of carbonate precipitation. Because modern surface-ocean P is regionally heterogeneous and influenced by differing biogeochemical conditions[44], it is important to ask whether our CAP records capture local or global changes in P. Absolute CAP values differ among sections, consistent with local differences in marine P concentrations across depositional settings (e.g., platform interior vs. slope, upwelling intensity, ventilation), but the timing and direction of CAP excursions are strikingly similar across all sites. In the modern ocean, local processes modulate P recycling, but are ultimately subordinate to the first-order control exerted by the deep-ocean P reservoir and its flux to the surface over geological timescales[21]. Multi-box marine P cycle models show that global changes in this reservoir can generate coherent CAP responses across diverse settings[32]. Under this framework, it is unlikely that independent local changes in P cycling on different parts of carbonate platforms or ramps would evolve in lockstep to produce the globally synchronous CAP trends we observe, in the absence of a global-scale perturbation to ocean P.

## Biogeochemical Modelling

Given that the CAP dataset presented here most likely represents near-primary trends resulting from changing ocean P levels across the LOME and LDME, using previously published redox proxy datasets allow us to empirically evaluate long-standing ideas on P cycling, anoxia, climate change and mass extinctions[10,38–41] (see Introduction). We therefore assess how these changing P levels affected major biogeochemical cycles and their impact on environmental and ecosystem conditions using the SCION biogeochemical model[45,46].

SCION is a 'predictive' forward biogeochemical model that uses boundary conditions, such as tectonic and palaeogeographic reconstructions, climate simulations, solid-Earth degassing and the timing of evolutionary events, whereas global surface chemistry, atmospheric $CO_2$ and climate emerge as model predictions (see Methods)[45]. We present and discuss model results for each mass extinction event (Fig. 3). In each model scenario, in order to simulate observed CAP peaks, we prescribe a set of P pulses to the ocean at intervals corresponding to the ages of the LOME and LDME. For the LOME model, we applied only one P pulse to the ocean, whereas for the LDME model, we applied two P pulses (Fig. 3a, g). This decision was made based on the observation of one CAP peak during the LOME and two CAP peaks during the LDME (Fig. 1). We then overlaid our CAP trends and associated geochemical data to evaluate the model predictions. We imposed a range of ocean P inputs, which are additional inputs above background P fluxes. These additional P inputs increase model oceanic P concentrations to various extents depending on the magnitude of the prescribed P pulse (Fig. 3a, b, g, h). A detailed description of the modelling can be found in the Methods section.

Our CAP data support an increase in ocean P levels starting near the initial LOME and LDME events and peaking towards the end of the extinction event (Fig. 1). Our model results show that the increase in marine P availability (Fig. 3b, h) triggers a surge in marine primary productivity, which causes a positive model $\delta^{13}C_{DIC}$ (DIC: dissolved inorganic carbon) excursion of ~1–18‰, depending on the P input flux (Fig. 3c, i). Proxies from the geological record support this increase in productivity, such as elevated $\delta^{13}C_{carb}$ (Fig. 3c, i), biogenic barium, total organic carbon[47], zinc isotopes[48], and pyrite sulphur isotopes[49–51]. Elevated rates of modelled marine productivity increase dissolved oxygen consumption in the water column via aerobic decomposition of this productivity (Fig. 3d, j). Negative shifts in $\delta^{238}U_{carb}$ trends point to an increase and/or variable bottom water anoxia during both the LOME and LDME (Fig. 3d, j)[7,8]. Both the LOME and LDME preserve widespread organic-rich shales[8,52], potentially linked to widespread increased productivity and ocean anoxia at that time[7,8]. The models show that the excess organic carbon burial, caused by increased P-limited productivity, increases atmospheric $O_2$ levels (Fig. 3e, k), and decreases atmospheric $CO_2$ levels, which in turn lowers global temperatures by up to around 5 °C (Fig. 3f, l). This model result aligns with carbonate and apatite $\delta^{18}O$ values in the LOME and LDME, that record 5–7 °C seawater temperature drops[5,7]. Note, however, that modelled temperature drops of ~5 °C are only attained in runs producing $\delta^{13}C_{carb}$ excursions of up to +18‰ which is much higher than those recorded during these extinction events (Fig. 3c, i). Discrepancies between the model and proxy temperature records may result from how sensitive the model results are to changing atmospheric $CO_2$ levels. The underlying palaeoclimate simulations in the SCION model have a low climate sensitivity, which does not allow large temperature deviations without substantially increasing, or decreasing, $CO_2$ levels[45]. Real-world climate sensitivity may be much higher. This may be evident from the significant differences between the model baseline temperatures (16–22 °C) and the $\delta^{18}O$-estimated baseline temperatures (30–35 °C) from the Ordovician and Devonian[6,37]. However, debate persists as to how accurate absolute temperature values are from oxygen isotope data[53].

## Role of P Spikes during the LOME and LDME

Despite the predictive model trends closely overlapping with our CAP data and previously published proxy data (Fig. 3), there are clear temporal differences between model predictions, observed geochemical proxies and extinction horizons, which calls for a reappraisal of the role that P played during these events (Fig. 4). Firstly, CAP data show that P levels were relatively low prior to and during the initial LOME extinction pulse, effectively ruling out P-driven anoxia as a cause for LOME1 (Figs. 1d,e; 4b), or alternatively suggesting that ecosystems

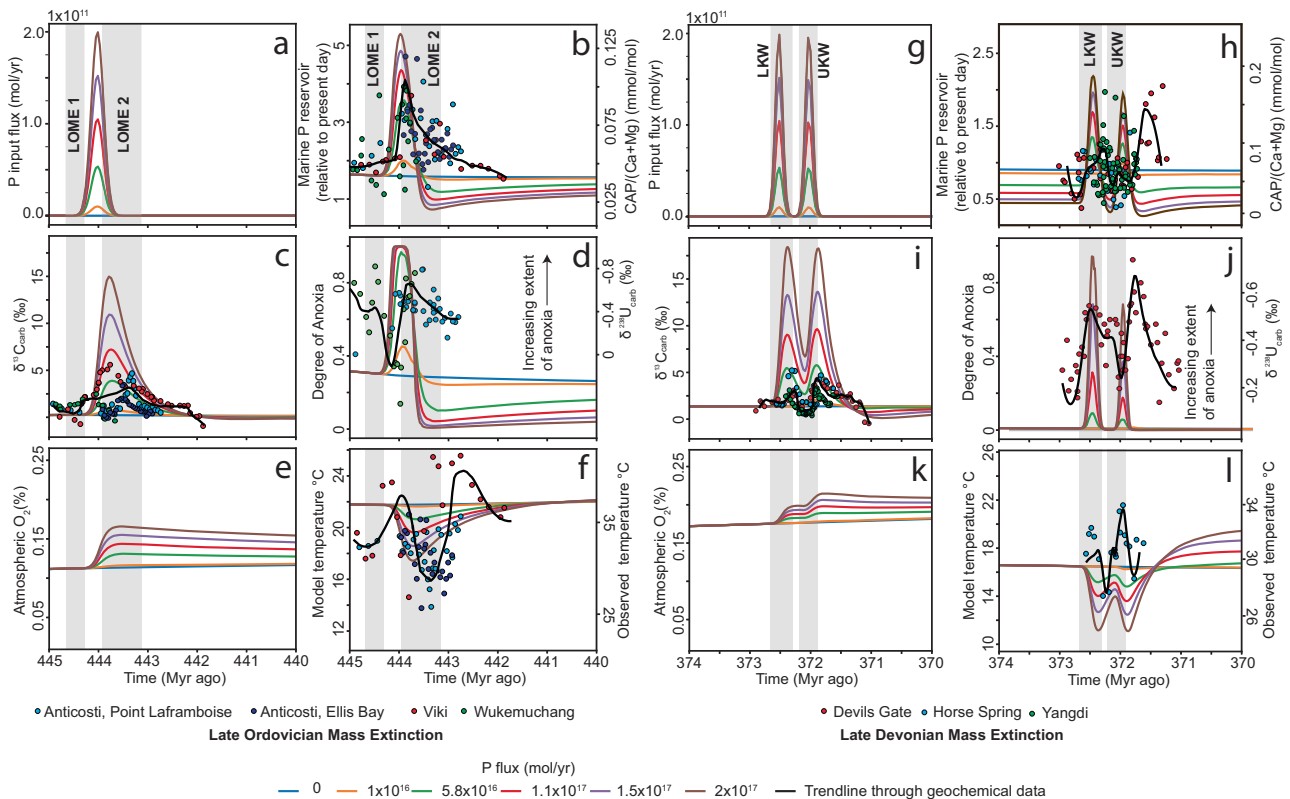

**Fig. 3 | Biogeochemical model results with overlain proxy data.** Subplots **a**–**f** and **g**–**l** correspond to the Late Ordovician and Devonian mass extinction events. **a**, **g** additional P input model forcing; **b**, **h** marine P concentration and CAP data (proxy for ocean P concentration); **c**, **i** carbon isotopic composition of marine dissolved inorganic carbon (DIC) and carbonate carbon isotope data (proxy for marine DIC composition); **d**, **j** marine uranium isotopic composition with carbonate hosted uranium isotopic composition (proxy for marine U composition); **e**, **k** atmospheric oxygen concentration; **f**, **l** model global temperature and observed temperature based on carbonate or phosphate oxygen isotopic composition. Black line shows trend line for proxy data. LOME 1 and 2, LKW and UKW correspond to previously defined extinction intervals (see text).

were especially sensitive to an initial increase in marine P levels and local increases in marine anoxia (cf. Fe-speciation vs. $\delta^{238}U_{carb}$ records)[7,18]. The former is supported by our model and $\delta^{238}U_{carb}$ data, which show that widespread marine anoxia does not occur until after LOME1 (Figs. 1b, d and 3d). This finding suggests that the LOME 1 extinction pulse may have been triggered by glacial cooling and sealevel fall[1], rather than eutrophication as a primary driver[11,12,17]. The subsequent marine P spike, aligning with the second LOME extinction pulse, coincides with a decline in $\delta^{238}U_{carb}$ values, which strongly implicates P-driven eutrophication and widespread anoxia as a critical kill mechanism (Fig. 4c)[1]. Interestingly, CAP spikes occur before the $\delta^{18}O_{carb}$ decrease, signifying that post-glacial weathering and warming are unlikely triggers for increasing marine P levels, but rather support P-driven cooling per the model results (Figs. 1b,e; 3f; 4c). Following the CAP spike, CAP values gradually decline throughout the Hirnantian and early Silurian, despite redox reconstructions based on U–Mo and Fe-speciation suggesting that extensive bottom-water anoxia continued well into the early Silurian[18,54,55], suggesting a more nuanced coupling between P and anoxia than a simple inverse relationship[56].

In contrast, the LDME shows a more intricate pattern. The first P pulse begins prior to and gradually increases across the LKW event, peaking just prior to the UKW extinction (Fig. 4e–f). This peak in CAP values, just prior to the UKW, may correlate with a spike in P accumulation rates in lake sediments during the LDME[57], this could support an increase in continental P delivery to the oceans. A temporally equivalent negative shift in $\delta^{238}U_{carb}$ moves in step with the increasing CAP values, which is then followed by a drop in CAP values and a positive shift in $\delta^{238}U_{carb}$ during the UKW extinction event (Figs. 3j, 4g). This antithetic relationship between CAP and $\delta^{238}U_{carb}$ supports the

notion that increasing global ocean P levels drove the expansion of seafloor anoxia and, in turn, contributed to extinction[57]. A second CAP peak occurring after the UKW extinction is again followed by another negative shift in $\delta^{238}U_{carb}$ (Fig. 3j), pointing towards another episode of P-driven marine anoxia, but during this second CAP peak, there is no extinction event recorded (Fig. 4h). While P-driven anoxia may not have resulted in an extinction, it may have prevented ecosystem recovery[58]. Interestingly, the CAP peaks during the LDME appear less distinct than those of the LOME (Fig. 1), which may reveal something about the magnitude or tempo of P fluxes during the event. For example, in our model, the P fluxes in both the LOME and LDME are of similar magnitude, but are of shorter duration during the LDME to match the reported age control[59]. These shorter duration pulses result in smaller changes to the global seawater P reservoir, which may result in CAP trends appearing less pronounced (Fig. 3h). Like the LOME, $\delta^{18}O_{carb}$ and $\delta^{18}O_{phosphate}$ data from the LDME show positive excursions interpreted as cooling events[5], that broadly overlap with the evolving antithetic CAP and $\delta^{238}U_{carb}$ relationship (Fig. 1g,h,i), supporting P-driven cooling via increased productivity and $CO_2$ drawdown (Fig. 4f,g,h). Global cooling in conjunction with marine anoxia would have exacerbated ecological stress and extinction.

Finally, we note that despite some mismatch in absolute values and durations, the modelling results and our empirical CAP data both support the same qualitative pattern: short-lived P pulses are capable of driving widespread anoxia, enhanced organic carbon burial, and global cooling. This convergence lends mechanistic support to P-driven extinction models[1,10–12,49,57,60,61] and strengthens the case for a potential causal link between transient P enrichment and environmental tipping points.

# Late Ordovician Mass Extinction        Late Devonian Mass Extinction

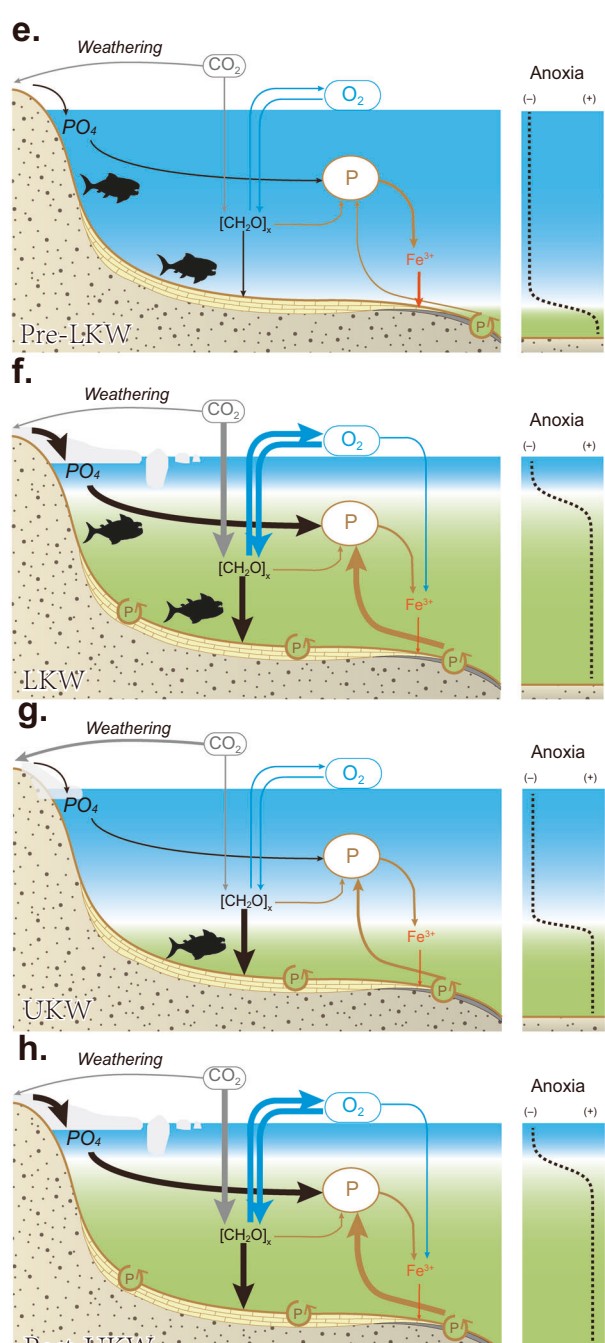

**Fig. 4 | Conceptual model for the LOME and LDME. a** Before LOME, weathering drew down atmospheric $CO_2$ levels and cooled the climate prior to the Hirnantian. **b** A change in ocean anoxia and/or cooling triggered the initial extinction pulse LOME 1. **c** An increase in continental P delivery to the oceans increased primary productivity, increasing dissolved oxygen demand and drawing down atmospheric $CO_2$ levels. Cooling and ocean anoxia resulted in widespread extinction. **d** Declining marine P levels reduced dissolved oxygen water column demand from decomposing primary productivity, and post-glacial warming allowed ecosystem recovery. **e** Before the LDME, the water column was mostly oxygenated under a greenhouse climate. **f** An increase in continental P delivery to the oceans increased primary productivity, increasing dissolved oxygen demand and drawing down atmospheric $CO_2$ levels. Cooling and ocean anoxia resulted in widespread extinction. **g** A reduction in marine P supply decreased P-driven productivity and reduced the extent of marine anoxia. **h** A second surge of P to marine environments increases the extent of marine anoxia and draws down atmospheric $CO_2$, culminating in global cooling, which together delayed ecosystem recovery.

## Phosphorus, Anoxia and Ecological Filtering Across the LOME and LDME

Our CAP records provide an opportunity to discuss ecological selectivity during both the LOME and LDME through a P-cycle lens, while also clarifying why ecological outcomes may have differed. In the Late Ordovician, the first extinction pulse (LOME 1) occurs when CAP values were low and there is minimal evidence for widespread anoxia. This suggests alternative extinction mechanisms such as global cooling, sea level regression and consequently shallow marine habitat loss, rather than primarily P-driven anoxia[1]. The LOME 1 extinction mainly affected

warm-adapted, environmentally restricted shelf faunas and some plankton in tropical epicontinental seas, whereas cool-water brachiopod assemblages (e.g., the Hirnantia fauna) expanded equatorward and many cosmopolitan taxa survived by tracking shifting depth and temperature belts[1,62,63]. In contrast, during LOME 2, when CAP rises and independent proxies indicate expanded anoxia, a wider range of taxa were affected, including both benthic and pelagic guilds. In this scenario, under expanded eutrophic, oxygen-poor shelf waters, plankton and nekton were squeezed into shrinking, oxygenated refugia, and smaller, deeper-water brachiopod associations dominated transgressive successions[62,64]. Given that warm, low-latitude surface waters hold less dissolved oxygen and tend to be more strongly stratified than cooler high-latitude and deeper waters, a short-lived P pulse would have promoted eutrophication and oxygen depletion most efficiently on tropical shelves, helping to explain why these settings show the highest extinction intensities during the P spikes[62,65]. After LOME-2, CAP records suggest marine P levels dropped to background levels, and complex, warm-water trophic webs reappeared locally in late Hirnantian nearshore settings, despite continuing shelf anoxia[1,66].

In the Late Devonian, by contrast, CAP and facies-dependent anoxia increased on low-latitude shelves at the time of the LKW (Fig. 1g-i), so the "first" Devonian pulse can be considered as a P-driven anoxia crisis for stromatoporoid–coral reefs and associated shallow-water benthos. Like the LOME, the most severe losses again focused on warm tropical shelves while high-latitude and deeper-water faunas were relatively buffered[5,67]. The CAP record shows a peak around the UKW, when reef builders, benthic invertebrates, and key planktonic and nektonic guilds suffered their most severe losses[59,67,68]. This can be explained by P-driven anoxia contracting habitable refugia to the point that even highly mobile nektonic clades could no longer escape hypoxic waters or maintain viable food webs. A secondary CAP peak in the earliest Famennian, without an extinction pulse, implies that sustained P-driven anoxia may have suppressed the recovery of reef, benthic and pelagic ecosystems, consistent with the prolonged "reef gap", and vertebrate size reductions observed in the post-Kellwasser world[58,59,68]. Taken together, these comparisons suggest a recurring, trait- and habitat-filtered sequence of ecological loss, with habitat loss first culling sessile and environmentally restricted benthic and reefal taxa. Sustained or rising peaks in marine P level and associated anoxia may then sufficiently erode remaining habitable environments, affecting mobile taxa as well, which culminates in broader extinction.

Our CAP dataset provides geochemical evidence that transient increases in marine P availability could have contributed to the expansion of ocean anoxia, global cooling, and biotic turnover during the first two major Phanerozoic mass extinction events, the LOME and LDME. While elevated marine P levels align with key extinction horizons in both the LOME and LDME, our data also suggest that some extinction pulses occurred under comparatively low P conditions. This highlights that P surges, while important, may not have been the sole driver of ecosystem collapses. This more nuanced picture challenges the prevailing assumption that elevated P levels invariably trigger mass mortality, and instead points to a more complex interplay among nutrient dynamics, redox feedbacks, climatic perturbations, and the ecological resilience of marine communities. Crucially, our study provides a globally integrated, quantitative reconstruction of seawater P concentrations across Phanerozoic major extinction intervals, offering geochemical evidence for the widely discussed, but previously undocumented changes in marine P levels thought to drive anoxia, climate change, and biotic crises. Moving forward, disentangling the roles of P from other co-occurring environmental changes and stressors will be essential to fully resolve the drivers of ancient oceanic extinction events and to better anticipate the risks posed by anthropogenic nutrient loading in the modern ocean.

## Method

### Carbon analysis
Total inorganic carbon (TIC) was measured using a FOGL Digital Soil Calcimeter. TIC was measured directly by weighing out ~0.2 g of rock powder into gas-tight 250 ml glass bottles and 4 ml of 6 M HCl in a plastic cuvette was carefully added to the bottle. After sealing, each bottle was inverted to initiate reaction between the carbonate powder and HCl, then intermittently agitated until the calcimeter reading stabilised. To monitor reproducibility, a pure calcium carbonate reference standard was run after every tenth sample, giving a precision of ±2% CaCO3 ($n = 30$).

### Carbonate carbon- and oxygen-isotope analyses
Carbonate carbon and oxygen isotope analysis was carried out at the West Australian Biogeochemistry Centre, University of Western Australia, and at Nanjing University. After drying under argon at 70 °C for 24 h, approximately 60-300 μg of sample powder was transferred to a vial. The dried powders were reacted under vacuum with 100% phosphoric acid at 70 °C using either a GasBench II or a Kiel IV system. The evolved $CO_2$ was then analysed isotopically on either a Thermo Fisher Scientific Delta XL mass spectrometer at the University of Western Australia or a MAT 253 instrument at Nanjing University. Isotope values were normalised against the IAEA reference materials NBS19 and NBS18 together with the Chinese national standard GBW04416. Carbonate $\delta^{13}C$ and $\delta^{18}O$ values are reported relative to VPDB, with precisions better than ±0.1‰ for carbon and ±0.15‰ for oxygen (1σ), based on duplicate analyses of standards and unknowns.

### Carbonate-associated phosphate (CAP) analysis
CAP was measured following previously established protocols[29], which is described below. Only samples containing more than 50% carbonate were included in the analysis. Sample masses were selected from TIC values and carbonate mineralogy so that each analysis contained approximately 1 mmol of carbonate. To remove adsorbed phosphorus, samples were washed repeatedly for 24 h with 4 mL aliquots of 10% NaCl buffered to pH 8 with $NaHCO_3$. After wash-solution phosphorus fell below 0.1 ppm, sufficient 2% vol/vol acetic acid was added to dissolve as much as 70% of the carbonate. After 30 min, the sample was centrifuged and the leachate was recovered through 0.2-μm polyethersulfone membrane filters. One leachate aliquot per sample was reserved for major- and trace-element analysis by Thermo Fisher Element XR ICP-MS at the Centre for Microscopy, Characterisation and Analysis, University of Western Australia. Elemental concentrations were reproducible to within ±2%, based on duplicate analyses of two internal standards. A separate leachate aliquot was used to determine phosphorus spectrophotometrically by the malachite green method at 663 nm on a Perkin-Elmer EnSight® plate reader at the University of Western Australia and Chengdu University of Technology; relative standard deviation was <±5%. The remaining residue was subsequently rinsed for 24 h with 4 mL of 10% NaCl buffered to pH 8 with $NaHCO_3$, and phosphorus in that wash was measured using the malachite green assay. The washing cycle was repeated until phosphorus in the wash solution fell below 0.1 ppm. The P in the leachate and washes were summed together to give CAP, which was then normalised to the Ca and Mg concentrations in the leachate as mmol of P per mol of Ca and Mg. For quality control, three Ediacaran dolostone materials spanning a range of TIC, TOC, and clay contents were analysed alongside unknowns, and duplicate CAP measurements differed by less than 10%.

### Biogeochemical modelling
We used the (py)SCION model[46], a pythonic version of SCION (Mills et al., 2021), to model changes in surface chemistry and temperature over the Palaeozoic. (py)SCION is a geological-timescale biogeochemical model that uses a precomputed climate emulator to resolve continental surface processes like weathering in 2D, and links these to a

nondimensional ocean-atmosphere that tracks concentrations of carbon, oxygen, sulfur, phosphorus and nitrogen compounds. For this work we modified the model P cycle after Longman et al. (2021) by adding an extra amount *Pforce* to the amount of bioavailable P entering the ocean in rivers during the model runs. For each pulse of P (one at the LOME and two at the LDME, note Longman et al. 2021 used two pulses for the LOME) we create a gaussian function (equation 1):

$$P_{LOME} \times 10^{-6} \times \frac{norm(t, -444, 0.5)}{norm(-444, -444, 0.5)} + P_{LDME1} \times 10^{-6}$$
$$\times \frac{norm(t, -373, 0.075)}{norm(-373, -373, 0.075)} + P_{LDME2} \times 10^{-6}$$
$$\times \frac{norm(t, -372, 0.075)}{norm(-372 -372, 0.075)} \quad (1)$$

Where $P_{LOME}$, $P_{LDME1}$ and $P_{LDME2}$ represent the total amount of bioavailable P for each event. The flux is also multiplied by a sedimentary P recycling factor (5x), which is otherwise not captured by *(py)SCION* but is observed to occur in more complicated models (in line with Longman et al. 2021). In our analysis we let $P_{LOME}$, $P_{LDME1}$ and $P_{LDME2}$ vary between $10^{16}$ and $2 \times 10^{17}$ in five regular intervals (as a reference point, the P flux values in Longman et al. 2021 are between 5 and $9 \times 10^{16}$).

## Data availability

The CAP, C and O isotope data generated in this study have been deposited in the Figshare database under accession code ZZ https://doi.org/10.6084/m9.figshare.30271312. Canning Basin phosphate oxygen isotope data are held in the UWA Repository https://research-repository.uwa.edu.au/en/persons/annette-george/datasets/.

## Materials availability

All samples were collected and exported in a responsible manner and in accordance with relevant permits and local laws. Global coordinates and/ or location information and drill core names are given for all samples collected in the Supplementary Information files. Requests for materials should be addressed to C.L., A.D., O.H., D.Y., X.Y., A.G., M.E., Q.W., C.B.

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

## Acknowledgements

C.L. acknowledges support from the NSFC (grants # 42425002) and the National Key Research and Development Program of China (grant numbers 2022YFF0800100). M.S.D. acknowledges support from the Forrest Research Foundation and UWA School of Earth and Oceans. B.J.W.M. is supported by UKRI grant EP/Y008790/1. A.D. acknowledges support from the research incubator of the Société du patrimoine Mondial Anticosti. O.H. was supported by the Estonian Research Council (grant PRG1701). A.S.M. is supported by ARC DECRA Fellowship DE230101642.

## Author contributions

M.S.D. and C.L. designed and organised the research. M.S.D., Z.Z., and A.Y.S. performed analyses. A.S.M. and B.J.W.M. performed modelling. C.L., A.D., O.H., D.Y., X.Y., A.G., M.E., D.W., Q.W., C.B. provided samples. The paper was written by M.S.D. with important inputs from all authors.

## Competing interests

The authors declare no competing interests.
