## [Peer Review file · Nature Communications]

Recurring Marine Phosphorus Spikes during Major Palaeozoic Mass Extinctions and Climate Change

Corresponding Author: Dr Matthew Dodd

Version 0:

Reviewer comments:

Reviewer #1

(Remarks to the Author)

Dodd et al. reported a line of new CAP datasets for several sections on different continents to track changes in oceanic P availability across biotic crisis horizons including the LOME and LDME. Combined bulk and in-situ determinations of CAP, to some extent, can avoid significant diagenetic overprints, which makes CAP more reliable for reconstructing seawater P levels of the past. The story is intriguing, especially with high-resolution and -quality data interpreted in a well-organized framework of data-modeling comparison. However, the current version may need moderate revisions to clarify several key questions to make it more novel before further consideration.

Major comments:

1. The recycling time of P in modern oceans is estimated one-two orders shorter than the mixing time, especially for the surface ocean (Paytan and McLaughlin, ACS, 2007). That means the P ratios associated with carbonate fractions may preserve more information of local marine environments. Moreover, modern ocean P cycle yields totally different influx and outflux for each reservoir, such as inshore, offshore, surface ocean and deep ocean that also responds differently to changes in terrestrial input and upwelling, also changes in sea level. Especially, during cooling intervals, enhanced ocean ventilation always induces more efficient recycling of P via upwelling. Under such scenario, upwelling contributed more than weathering input to primary production enhancement either in rate or in extent (Slomp and Van Cappellen, BG, 2007). So, did this paper consider potential difference in CAP trends generated for those carbonates formed in different locations of carbonate platform or ramp? And, can the P cycle as described in the SCION differentiate the contributions of weathering input and upwelling? See Chi Fru et al. (NC, 2023) as a good analog.
2. Increased oceanic P availability, as here indicated by CAP spikes, is surely a key chain of the climate-ocean feedback loop yet always the intermediate one. This hypothesis does provide us important clues about how the tectonic and/or climatic shifts caused oceanic changes (e.g., eutrophication / deoxygenation) but can't be simply summarized as a trigger. After all, some of previous works argue for a more important role of cooling in initiating biotic crisis. For example, LOME may have been triggered by cooling (or earlier before the first episode) rather than subsequent more extensive bottom-water anoxia during the deglacial (Harper, NSR, 2023). Instead, recurrent P spikes during the LOME and LDME do suggest similarity in the mechanisms linking climatic cooling to oceanic anoxia, which is certainly a strike finding for this study. So, "triggered" may be not preferred for the geobiology community.
3. Regarding ocean redox state in the Early Silurian, this study, as concluded in the final cartoon, seems inconsistent with previous papers (Zou et al., 2018; Stockey et al., NC, 2021; Young et al., PPP, 2021; Qiu et al., CEE, 2022; Zhang et al., EPSL, 2024). As I mentioned above, CAP decrease in shallow shelf environments may be not direct evidence for ocean oligotrophication. Both Mo and U isotopes indicate an expansion of bottom-water anoxia since the end of glaciation (Dahl et al., ESR, 2021; Stockey et al., NC, 2021). For this stage, CAP may be still a key yet not a control.

Minor comments:

Lines 57-58: it's still under debate whether there is a LIP associated with the LOME, except that condense volcanic ash layer identified around the LOME horizon.

Line 71: "P-driven" is confusing.

Lines 74-88: It seems bulk CAP data were not reported, for the first time, for the WUKE section (Zhang et al., EPSL, 2024). Actually, WUKE is right WUKEMUCHANG (two names for the same section).

Lines 249-258: The SCION is originated from the COPSE, in which a low climate sensitivity was used. More importantly, biogeochemical models are always insufficient to constrain the interactions between climate and CO₂ levels due to the lack

of ocean physics module. For example, the tipping point for temperature falls and glaciation is usually hard to constrain for most biogeochemical models.

Fig.1: There was sedimentary hiatus and/or surface erosion between the Ellis Bay Formation and Becscie Formation for the EB section, Anticosti. Thus, the boundary line should be dotted. And it's similar for the Wukemuchang section, because graptolites from carbonate successions remains uncertain when used to constrain stage boundary.

Fig.4: As mentioned in the 3rd comment, panel d in the cartoon may need modification. P.S., fish was popular for the LDME but was very rare for the LOME.

Reviewer #2

(Remarks to the Author)

This is an innovative and well-written study. Although numerous independent geochemical and modeling works have examined the Late Ordovician (LOME) and Late Devonian (LDME) mass extinctions separately, Dodd & Li et al. provide a novel and integrative approach by comparing these two intervals through the lens of carbonate-associated phosphate (CAP).

By isolating the seawater phosphate signal from other phosphorus phases, the authors convincingly demonstrate a synchronous relationship between elevated marine phosphate concentrations and oceanic anoxia. They further validate that CAP reliably records primary seawater signals rather than diagenetic or secondary overprints. The subsequent numerical modeling under different boundary conditions elegantly links variations in productivity, anoxia, and carbon-cycle perturbations across both extinction intervals. The use of CAP as a paleo-phosphate proxy is both frontier and original, and this paper makes an important conceptual contribution. I recommend minor revision before acceptance.

A few suggestions for the authors, most comments should not be viewed as being particularly critical but rather suggestions and helpful "food for thought":

The use of CAP as a proxy is highly innovative and forward-looking. I think the Introduction could briefly expand on how dissolved phosphate increases in seawater, which would help a broader audience quickly follow the authors' reasoning. I am particularly interested in the relationships among different phosphorus pools and how they interconvert—how do particulate or organic P species transform into dissolved phosphate, and under what conditions?

In anoxic settings, organic-rich black shales can release phosphorus from organic matter back into the water column. During intervals like the second pulse of the LOME and the LDME—both marked by globally widespread black shale deposition—could such P regeneration have provided a positive feedback, enhancing marine phosphate levels while also stimulating productivity? And could this regenerated phosphorus have dissolved into seawater and been captured by CAP? (See, for instance, the early classic work by Ingall et al., GCA, 1993.)

I also wonder whether global seawater phosphate concentrations were spatially homogeneous. Could there have been regional heterogeneity in dissolved phosphate? According to an Earth system model (Liu et al., 2025, ESR), surface chlorophyll concentrations—an even more direct proxy for productivity—rose substantially during the O–S transition but were distributed heterogeneously, with higher values at high latitudes. If these proxies are potentially linked, the finding would be even more compelling.

Additionally, although CAP trends remain broadly consistent among different sections, their absolute values variable. What might cause these differences? Local redox gradients, productivity intensity, or minor diagenetic effects?

Finally, the paper's impact could be strengthened if the Implications section discussed more explicitly how changes in productivity and anoxia affected different ecological groups. For example, LOME-1 primarily affected benthic fauna, whereas LOME-2 had a broader ecological reach. Similarly, during the LDME, the Lower Kellwasser Event had a stronger effect on reef builders such as corals, while the Upper Kellwasser Event was more severe and also influenced nektonic organisms. Expanding on these ecological contrasts would help build a closer connection between phosphorus cycling and biotic evolution—and would make the paper even more engaging for a wider readership.

Line-line comments

Line 59: The phrase "land plant-related weathering" fits LDME but is unlikely relevant to LOME. Please clarify this distinction.

Line 62: Zhang Junpeng et al. (2024, EPSL) presented CAP data from the Wuke carbonate section that support their numerical model. The statement "no geochemical evidence that marine phosphorus levels..." is therefore too absolute and should be softened.

Line 82: The first appearance of any biozone name in the main text should use its full form rather than abbreviations.

Lines 158–160: The relatively low $\delta^{18}\text{O}$ values of the Wuke carbonates likely reflect meteoric water influence, which has extremely low phosphate content and would not alter the CAP composition. Reference Liu et al. (2022, EPSL) could be cited here to strengthen the discussion on diagenetic effects.

Line 227: For the LDME, the two CAP peaks mentioned in the text appear less distinct. Please clarify or quantify this observation.

Line 262: Minor discrepancies between the modeled and observed records are acceptable, considering the substantial stratigraphic correlation uncertainties across different paleocontinents.

In Fig. 1B, the rightmost panel lacks a label identifying the geochemical proxy. Fonts appear distorted and should be standardized. Biostratigraphic names should be italicized. The lithologic column needs a legend to explain lithology symbols.

REVIEWER COMMENTS

Reviewer #1 (Remarks to the Author):

Dodd et al. reported a line of new CAP datasets for several sections on different continents to track changes in oceanic P availability across biotic crisis horizons including the LOME and LDME. Combined bulk and in-situ determinations of CAP, to some extent, can avoid significant diagenetic overprints, which makes CAP more reliable for reconstructing seawater P levels of the past. The story is intriguing, especially with high –resolution and –quality data interpreted in a well-organized framework of data-modeling comparison. However, the current version may need moderate revisions to clarify several key questions to make it more novel before further consideration.

Major comments:

1. The recycling time of P in modern oceans is estimated one-two orders shorter than the mixing time, especially for the surface ocean (Paytan and McLaughlin, ACS, 2007). That means the P ratios associated with carbonate fractions may preserve more information of local marine environments. Moreover, modern ocean P cycle yields totally different influx and outflux for each reservoir, such as inshore, offshore, surface ocean and deep ocean that also responds differently to changes in terrestrial input and upwelling, also changes in sea level. Especially, during cooling intervals, enhanced ocean ventilation always induces more efficient recycling of P via upwelling. Under such scenario, upwelling contributed more than weathering input to primary production enhancement either in rate or in extent (Slomp and Van Cappellen, BG, 2007). So, did this paper consider potential difference in CAP trends generated for those carbonates formed in different locations of carbonate platform or ramp? And, can the P cycle as described in the SCION differentiate the contributions of weathering input and upwelling? See Chi Fru et al. (NC, 2023) as a good analog.

We thank the reviewer for their time and feedback to help improve the manuscript.

Yes, we agree that CAP values will represent local P concentrations, and indeed ocean surface waters are heterogenous with respect to P concentrations, meaning absolute CAP values are not faithful records of average surface seawater concentrations. We instead draw conclusions based on the CAP trends rather than their absolute values. As seen in figure 2, sections from globally disparate locations and environmental settings preserve similar CAP trends suggesting that while P concentrations may have varied among sites the global ocean P reservoir likely changed in accordance with the trends recorded by CAP. This is an important point which has been raised by reviewer 2 as well and we have therefore added text on lines 225-237 to address this key point.

The SCION model is not able to represent changes in upwelling explicitly as it has a single-box ocean. However, given we are investigating overall ocean P inventory change over millions of years, and upwelling operates on thousands of years, we expect changing upwelling rates to have relatively little effect on this overall P inventory, although they could change P distribution as above. The model used in Chi Fru et al., 2023, has multiple ocean boxes and includes water mass fluxes which allows one to change the rate of upwelling, but in that paper it was used to explore the effects of P inputs into different ocean environments rather than changes in upwelling rates. We have clarified these points in the revision.

2. Increased oceanic P availability, as here indicated by CAP spikes, is surely a key chain of the climate-ocean feedback loop yet always the intermediate one. This hypothesis does provide us important clues about how the tectonic and/or climatic shifts caused oceanic changes (e.g., eutrophication / deoxygenation) but can't be simply summarized as a trigger. After all, some of previous works argue for a more important role of cooling in initiating biotic crisis. For example, LOME may have been triggered by cooling (or earlier before the first episode) rather than subsequent more extensive bottom-water anoxia during the deglacial (Harper, NSR, 2023). Instead, recurrent P spikes during the LOME and LDME do suggest similarity in the mechanisms linking climatic cooling to oceanic anoxia, which is certainly a strike finding for this study. So, "triggered" may be not preferred for the geobiology community.

We agree that the term ‘trigger’ should be more carefully used as the P spikes themselves can of course be triggered by other events, so we have reworded the title and modified the text on lines 377-379 to refer to marine P spikes as a key feature rather than the initial trigger of events.

3. Regarding ocean redox state in the Early Silurian, this study, as concluded in the final cartoon, seems inconsistent with previous papers (Zou et al., 2018; Stockey et al., NC, 2021; Young et al., PPP, 2021; Qiu et al., CEE, 2022; Zhang et al., EPSL, 2024). As I mentioned above, CAP decrease in shallow shelf environments may be not direct evidence for ocean oligotrophication. Both Mo and U isotopes indicate an expansion of bottom-water anoxia since the end of glaciation (Dahl et al., ESR, 2021; Stockey et al., NC, 2021). For this stage, CAP may be still a key yet not a control.

The cited papers do indeed support continued extensive degrees of bottom-water anoxia in the early Silurian following the LOME, which does contrast somewhat with the decline in CAP values in all sections post LOME. This may suggest that additional factors such as weathering, not anoxia alone sustained higher marine P levels during LOME 2.

Minor comments:

Lines 57-58: it's still under debate whether there is a LIP associated with the LOME, except that condense volcanic ash layer identified around the LOME horizon.

We have modified this sentence to include the possibility of LIP in these events as follows:” These two extinction events differ from the remaining “big five” extinctions, in that they correlate with global cooling, as opposed to global warming³, yet they may still have been linked with large igneous provinces and associated volcanic activity⁴.

Line 71: “P-driven” is confusing.

Changed sentence to: “As a result, the proposed role of phosphorus in these extinctions remains speculative and poorly constrained”.

Lines 74-88: It seems bulk CAP data were not reported, for the first time, for the WUKE section (Zhang et al., EPSL, 2024). Actually, WUKE is right WUKEMUCHANG (two names for the same section).

We thank the reviewer for bringing this to our attention. The section analysed in the cited paper is the same as this study and reports a similar CAP trend. We now cite this paper and have removed the phrasing “for the first time” when referencing that section.

Lines 249-258: The SCION is originated from the COPSE, in which a low climate sensitivity was used. More importantly, biogeochemical models are always insufficient to constrain the interactions between climate and CO₂ levels due to the lack of ocean physics module. For example, the tipping point for temperature falls and glaciation is usually hard to constrain for most biogeochemical models.

We have added a brief statement on lines 278-284 to highlight this issue with biogeochemical models when reconstructing climate change.

Fig.1: There was sedimentary hiatus and/or surface erosion between the Ellis Bay Formation and Becscie Formation for the EB section, Anticosti. Thus, the boundary line should be dotted. And it's similar for the Wukemuchang section, because graptolites from carbonate successions remains uncertain when used to constrain stage boundary.

We have modified the geological columns to include the hiatus.

Fig.4: As mentioned in the 3rd comment, panel d in the cartoon may need modification. P.S., fish was popular for the LDME but was very rare for the LOME.

We have modified the cartoon to reflect expansion of anoxia post LOME per the 3rd comment,

and we liked the suggestion to use fish in the LDME and have modified accordingly.

Reviewer #2 (Remarks to the Author):

This is an innovative and well-written study. Although numerous independent geochemical and modeling works have examined the Late Ordovician (LOME) and Late Devonian (LDME) mass extinctions separately, Dodd & Li et al. provide a novel and integrative approach by comparing these two intervals through the lens of carbonate-associated phosphate (CAP).

By isolating the seawater phosphate signal from other phosphorus phases, the authors convincingly demonstrate a synchronous relationship between elevated marine phosphate concentrations and oceanic anoxia. They further validate that CAP reliably records primary seawater signals rather than diagenetic or secondary overprints. The subsequent numerical modeling under different boundary conditions elegantly links variations in productivity, anoxia, and carbon-cycle perturbations across both extinction intervals. The use of CAP as a paleo-phosphate proxy is both frontier and original, and this paper makes an important conceptual contribution. I recommend minor revision before acceptance.

A few suggestions for the authors, most comments should not be viewed as being particularly critical but rather suggestions and helpful “food for thought”:

We thank the reviewer for their time and suggestions to improve the manuscript.

The use of CAP as a proxy is highly innovative and forward-looking. I think the Introduction could briefly expand on how dissolved phosphate increases in seawater, which would help a broader audience quickly follow the authors’ reasoning. I am particularly interested in the relationships among different phosphorus pools and how they interconvert—how do particulate or organic P species transform into dissolved phosphate, and under what conditions?

We have added a new paragraph to the introduction section to describe how P is cycled within the oceans and sediments.

In anoxic settings, organic-rich black shales can release phosphorus from organic matter back into the water column. During intervals like the second pulse of the LOME and the LDME—both marked by globally widespread black shale deposition—could such P regeneration have provided a positive feedback, enhancing marine phosphate levels while also stimulating productivity? And could this regenerated phosphorus have dissolved into seawater and been captured by CAP? (See, for instance, the early classic work by Ingall et al., GCA, 1993.)

Yes, the correlation of proxies for bottom-water anoxia (d238U) and CAP in the LOME and LDME support a positive feedback between anoxia and marine P levels and is described on lines 348-350.

I also wonder whether global seawater phosphate concentrations were spatially homogeneous. Could there have been regional heterogeneity in dissolved phosphate? According to an Earth system model (Liu et al., 2025, ESR), surface chlorophyll concentrations—an even more direct proxy for productivity—rose substantially during the O–S transition but were distributed heterogeneously, with higher values at high latitudes. If these proxies are potentially linked, the finding would be even more compelling.

Global surface seawater phosphate concentrations were likely spatially heterogeneous like in the modern surface ocean layers. As discussed in comment 1.1 we drew conclusions from CAP trends rather than absolute CAP values among sections. Meaning we analysed how seawater P levels changed not their relative concentration across the globe. Nevertheless, the modelled chlorophyll concentrations in Liu et al., 2025, represent an increase in productivity which would be supported by elevated seawater P levels as supported by CAP during the O-S transition. We have added the citation Liu et al, 2025 in our discussion on lines 181 as a linkage between the chlorophyll model productivity prediction and the CAP data.

Additionally, although CAP trends remain broadly consistent among different sections, their absolute

values variable. What might cause these differences? Local redox gradients, productivity intensity, or minor diagenetic effects?

All of the above may have contributed to differences in local P concentrations, which likely explains the differences in absolute CAP values among sections. Consequently, we used a minimum of 3 globally distributed sections in this study for each extinction event in order to best reflect global seawater changes in marine P levels not just local changes. We have included and extended discussion on this in lines 225-237.

Finally, the paper's impact could be strengthened if the Implications section discussed more explicitly how changes in productivity and anoxia affected different ecological groups. For example, LOME-1 primarily affected benthic fauna, whereas LOME-2 had a broader ecological reach. Similarly, during the LDME, the Lower Kellwasser Event had a stronger effect on reef builders such as corals, while the Upper Kellwasser Event was more severe and also influenced nektonic organisms. Expanding on these ecological contrasts would help build a closer connection between phosphorus cycling and biotic evolution—and would make the paper even more engaging for a wider readership.

We agree this will improve the papers impact and have included a detailed discussion on lines 336-371.

Line-line comments

Line 59: The phrase “land plant–related weathering” fits LDME but is unlikely relevant to LOME. Please clarify this distinction.

We have added a bracketed comment to state this is relevant primarily to the LDME on line 60.

Line 62: Zhang Junpeng et al. (2024, EPSL) presented CAP data from the Wuke carbonate section that support their numerical model. The statement “no geochemical evidence that marine phosphorus levels...” is therefore too absolute and should be softened.

We thank the reviewer for bringing this to our attention. The section analysed in the cited paper is the same as this study and reports a similar CAP trend. We now cite this paper and have removed the phrasing “for the first time” when referencing that section.

Line 82: The first appearance of any biozone name in the main text should use its full form rather than abbreviations.

Full names are now presented in the first instances in the text.

Lines 158–160: The relatively low $\delta^{18}\text{O}$ values of the Wuke carbonates likely reflect meteoric water influence, which has extremely low phosphate content and would not alter the CAP composition. Reference Liu et al. (2022, EPSL) could be cited here to strengthen the discussion on diagenetic effects.

We agree the low $\delta^{18}\text{O}$ values of the Wuke carbonates likely reflect diagenetic alteration. Despite this CAP and $\delta^{238}\text{U}$ data remain consistent between globally distributed sections, suggesting that while some easily altered features of carbonate may be altered other geochemical aspects can retain near-primary signatures. We have included this reference in the discussion on diagenetic effects.

Line 227: For the LDME, the two CAP peaks mentioned in the text appear less distinct. Please clarify or quantify this observation.

The relative magnitude of the CAP peaks may reflect the duration and/or magnitude of the P flux during the LDME relative to the LOME. For example, in our SCION experiments we impose a shorter-duration pair of P pulses for the LKW and UKW than for the Hirnantian event, due to the reported event timelines. This results in a smaller net increase in the global marine P reservoir during the LDME relative to the LOME despite, a similar P flux magnitude in both events. Consequently, the relative change in CAP during both events may speak to the magnitude of P fluxes involved. This discussion has been added to lines: 319-324.

Line 262: Minor discrepancies between the modeled and observed records are acceptable, considering the substantial stratigraphic correlation uncertainties across different paleocontinents. In Fig. 1B, the rightmost panel lacks a label identifying the geochemical proxy. Fonts appear distorted and should be standardized. Biostratigraphic names should be italicized. The lithologic column needs a legend to explain lithology symbols.

Thank you for spotting these figure label and formatting issues. All has been addressed in the revised figure 1.